# Impact ionization by hot carriers in a black phosphorus field effect transistor

Faisal Ahmed[1,2], Young Duck Kim[3,4], Zheng Yang[2], Pan He[5], Euyheon Hwang[2], Hyunsoo Yang [5], James Hone[3] & Won Jong Yoo [1,2]

The strong Coulombic interactions in miniaturized structures can lead to efficient carrier multiplication, which is essential for many-body physics and design of efficient photonic devices beyond thermodynamic conversion limits. However, carrier multiplication has rarely been realized in layered semiconducting materials despite strong electronic interactions. Here, we report the experimental observation of unusual carrier multiplication in a multilayer black phosphorus device. Electric field-dependent Hall measurements confirm a substantial increase of carrier density in multilayer black phosphorus channel, which is attributed to the impact ionization by energetic carriers. This mechanism relies on the generation of self-heating induced charge carriers under the large electric field due to competition between electron–electron and electron–phonon interactions in the direct and narrow band gap (0.3 eV) of the multilayer black phosphorus. These findings point the way toward utilization of carrier multiplication to enhance the performance of electronics and optoelectronics devices based on two-dimensional materials.

[1] School of Mechanical Engineering, Sungkyunkwan University, 2066, Seobu-ro, Jangan-gu, Suwon, Gyeonggi-do 16419, Korea. [2] SKKU Advanced Institute of Nano-Technology (SAINT), Sungkyunkwan University, 2066, Seobu-ro, Jangan-gu, Suwon, Gyeonggi-do 16419, Korea. [3] Department of Mechanical Engineering, Columbia University, New York, NY 10027, USA. [4] Department of Physics, Kyung Hee University, Seoul 02447, Korea. [5] Department of Electrical and Computer Engineering, and NUSNNI, National University of Singapore, Singapore 117576, Singapore. Correspondence and requests for materials should be addressed to W.J.Y. (email: yoowj@skku.edu)

mpact ionization is a carrier multiplication (CM) process by which more electron-hole pairs are generated due to strong Coulombic interactions between charge carriers. Impact ionization is a three-particle down-conversion process in which a high-energy (hot) conduction band electron (or valance band hole) interacts with a valence band electron and excites the electron across the energy gap, leaving behind a hole[1–5]. The process is also called an inverse Auger recombination process[3,5]. The high kinetic energy of the carriers excites electron-hole pairs prior to dissipation via scattering, mainly optical phonon scattering. Impact ionization is important for realizing ultrafast and energy efficient optoelectronic devices, such as solar cells, photodetectors, and electroluminescent emitters[1–4]. Impact ionization has been studied in bulk semiconductors[6], and has been observed in several miniaturized structures as the relaxation dynamics are strongly modified with scaling via quantum confinement effects[7–9]. The efficient CM can be expected in a semiconducting device, when low dielectric screening and low doping conditions are realized that results in strong electron-electron interactions. Therefore, CM may be readily achieved in layered two-dimensional (2D) materials in which the charge confinement and low dielectric screening due to atomically thin structure give rise to strong Coulombic interactions[4,9,10]. However, most of the commonly studied layered semiconducting materials, such as transition metal dichalcogenides (TMDCs), possess a wide band gap (1–2 eV), a low carrier mobility (several tens of $cm^2V^{-1}s^{-1}$), and a small optical phonon energy[9]. Taken together, these factors hinder the observation of efficient CM in these systems. Herein, we report the realization of efficient CM due to impact ionization by hot carriers in multilayer black Phosphorus (BP) devices, which feature a narrow and direct band gap (0.3 eV) and a high carrier mobility[11].

BP is a nascent layered semiconducting material and stands out due to its unique puckered honeycomb lattice structure, which induces strong anisotropy along its in-plane directions[12]. BP exhibits strong carrier confinement in the out-of-plane direction; therefore, the thickness may be used to sensitively modulate its physical properties. For example, the energy gap of BP is 2 eV in a monolayer and 0.3 eV in a multilayer structure (over 6 layers), covering a broad electromagnetic spectrum[9]. The charge carriers of BP display mobility values of several hundred $cm^2V^{-1}s^{-1}$ at room temperature, and these values are further increased by dielectric engineering and reducing the temperature to cryostat conditions. As a result, BP displays quantum oscillations at low temperatures in the presence of a magnetic field that is rarely observed in other layered semiconducting materials in their pristine form[13]. Although these studies are highly useful for understanding the inherent material properties and device physics, the practical utility of these devices is limited: devices are operated under harsh environments, such as high electrical fields and operating temperatures. In an effort to integrate BP into devices, we examined the carrier transport properties under a large electric field.

Usually, semiconducting devices operated under large electric fields exhibit a current saturation due to electrostatic channel pinch-off or to velocity saturation of the charge carriers[14]. Current saturating behavior in layered materials, however, has been elusive due to the thin geometry, therefore, most of the layered semiconducting materials exhibit a soft saturating behavior in very thin high dielectric constant ($k$) insulating environments. Several groups have reported current saturation in multilayer BP devices[11,15,16], to explore its potential use in radio frequency applications. However, the saturating behavior in few layer BP devices was only observed in high-$k$ and/or thin dielectric environments that induce very strong screening effect (see the Supplementary Note 1 and Supplementary Table 1).

Here, we fabricate several back-gated multilayered BP devices supported on a 285 nm $SiO_2$ substrate with few-layered $h$BN that induce relatively weak charge screening effect. A linearly increasing current level under an electric field, referred to herein as super-linear behavior, is observed instead of saturation. Engel et al. realized a similar trend experimentally[17], and Trushkov et al. very recently predicted this behavior theoretically[18]. These reports, along with our experimental results suggest the presence of an alternative carrier transport mechanism at high electrical field values. We attribute the origin of the super-linear behavior to CM due to impact ionization by hot carriers in the BP channel under low dielectric screening conditions, because the generated charge carriers appear to overwhelm current saturation at high electric fields. Other possible transport mechanisms such as the large contact resistance and hopping transport are systematically considered. To experimentally test whether CM by impact ionization occurred in this system, we compute the exact number of charge carriers, i.e., the Hall carrier density ($n_H$), in $h$BN-encapsulated BP devices as a function of the electric field. Interestingly, we observe a sudden increase in the $n_H$ of BP at high electric fields, strongly suggesting the presence of CM in the channel. The increased number of charge carriers affect the potential distribution, as evidenced by the change in the capacitive coupling (CC) between the BP channel and the back-gate dielectric. This study will be helpful for realizing practical and energy efficient low-power optoelectronic devices.

## Results

**Low electrical field characterization of BP.** We fabricated $h$BN-encapsulated BP devices as shown schematically in Fig. 1a, considering the environmentally sensitive nature of BP[19,20]. The bottom $h$BN layers protected the BP channel from oxide-trapped charges that introduce hysteresis (see the Supplementary Figure 1), whereas the top $h$BN layers avoided unwanted oxidation and hydration of BP. The results presented below were obtained from $h$BN-encapsulated BP devices unless otherwise noted. The exfoliation and transfer of BP were carried out in the controlled inert environment of a glove box with less than 1 ppm moisture and oxygen. During fabrication, the candidate flakes were selected by optical contrast, and Raman spectra were acquired to ascertain the chemical identity of materials (see the Supplementary Figure 2 and Supplementary Note 2). Soon after fabrication, the devices were measured under vacuum conditions at room temperature. The measured low electrical field results were compiled in Fig. 1b as 2D color plots of the source-to-drain current ($I_D$) as a function of the applied drain ($V_D$) and gate ($V_G$) bias values. These results were obtained from 24 nm thick BP device encapsulated with 20 nm top and 26 nm thick bottom $h$BN flakes, device (1), (see the Supplementary Figure 3 and Supplementary Table 2). The current level increased as $V_D$ decreased towards more negative values for both positive and negative $V_G$ regions. This behavior depicted a slight ambipolar nature of BP with a large hole current appearing towards the negative $V_G$ side and small electron current towards the positive $V_G$ side. As more negative (positive) $V_G$ was applied, the BP bands are bent upwards (downwards), thereby decreasing the barrier height and width for holes (electrons) along the BP–metal interface, as shown in their respective energy band diagrams. This effect facilitated carrier injection into the channel and increased the current level. This trend confirmed the p-dominated ambipolar nature of the multilayer BP device[16].

**High electrical field response of BP.** Next, we applied a large electric field to the BP devices to study the electrical endurance and carrier transport in extreme operating conditions. This step

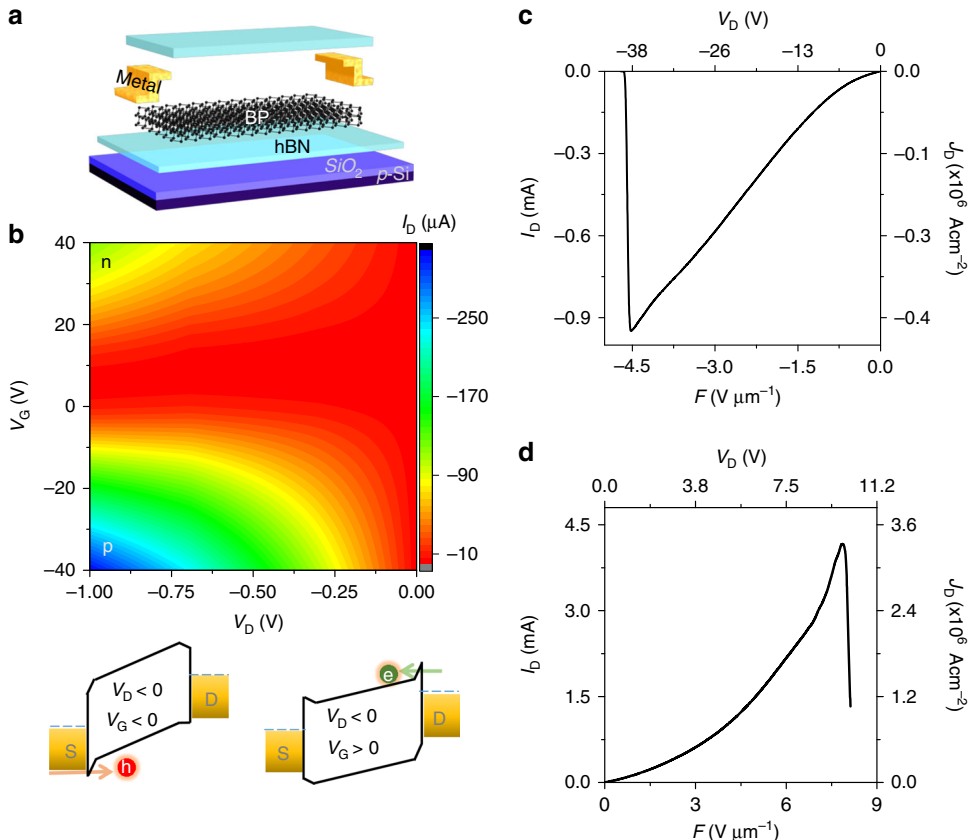

**Fig. 1** Characterization of a multilayer BP device and its breakdown. **a** Schematic diagram illustrating the fabrication steps of a back-gated $h$BN-encapsulated BP FET device. **b** 2D contour plot representing the device current ($I_D$) as a function of various $V_D$ (from 0 to −1 V) and $V_G$ (from 40 to −40 V), where n and p indicate electron and hole branches and their corresponding band alignment positions of the BP channel at given bias conditions. Here, green and red circles are electron and hole carriers respectively. **c** High electric field Joule breakdown at $V_G = 0$ towards negative field region in the BP device (1) (**d**) Joule breakdown towards positive field region in BP device (2)

was accomplished by slowly increasing the maximum applied electric field until the Joule breakdown occurred. Our obtained results from BP device (1) showed increasing current level with applied negative $V_D$, as shown in Fig. 1c. Similarly, we measured another BP device (2) towards the positive $V_D$ region (see the Supplementary Figure 4). Interestingly, it also showed similar increasing current-voltage ($I$–$V$) trend, as shown in Fig. 1d. It confirmed that the obtained super-linear trend was independent of $V_D$ sweep direction. However, the difference in the obtained current levels in Fig. 1c, d was due to different thickness of flakes used, different channel parameters and mainly different nature of band alignments in $p$-dominated BP channel by applying very large forward and reverse drain bias, as explained by Das et al. previously[16]. We repeated the experiments to the SiO$_2$ and/or $h$BN supported back-gated BP devices of more than 30 different thicknesses (6 to 42 nm thick BP flakes), and all the measured devices displayed a super-linear trend indicating that the observed trend was reproducible, (see the Supplementary Figures 5-7). Interestingly, this trend continues up to the breakdown point, at which time a sudden dip in the current level was observed due to Joule breakdown. The nature of breakdown is discussed in the last part of the manuscript. In view of the fact that BP is an anisotropic material due to its puckered geometry[12], we fabricated several BP devices along different in-plane directions. No sign of current saturation was observed (see the Supplementary Figure 7). It is possible that an alternative carrier transport mechanism is present and capable of masking current saturation in BP, as explained subsequently.

The observed increase in the current level with the applied electric field in BP devices might be due to (i) the large contact resistance ($R_c$) across the metal–BP contact, (ii) hopping transport in the channel, or (iii) CM due to impact ionization by hot carriers. The mechanism underlying the super-linear trend was sought by exploring the effects of these three phenomena on the BP device. The interfacial contact of layered semiconducting materials and metallic electrodes is typically of a Schottky nature, with large $R_c$ due to strong Fermi level pinning effects, wide band gap and weak bonding to the metal. In such devices, the overall device performance is dominated by the contacts rather than the channel[21,22], with charge injection either by thermionic emission over the barrier and/or tunneling across them[23]. Both mechanisms can show decreased resistance at high fields, due to Joule heating (thermionic emission) and barrier narrowing (tunneling). We observed linear output curves in all of our measured devices, as shown in Fig. 2a, that suggest a small induced $R_c$ and formation of Ohmic-like interfacial barriers. This was further confirmed by carrying out four-probe measurements to obtain $R_c$ of the Cr-BP contact, using the equation (1),

$$2R_c = (R_{2P} - R_{4P}) \times (L/W) \tag{1}$$

where, $R_{2P}$ and $R_{4P}$ are two-probe and four-probe resistances, respectively, $L$ is length and $W$ is width of BP channel. The extracted $R_c$ value fell within the range of several kΩ, as shown in inset of Fig. 2b, that is merely 10% of total device resistance at $V_G = -25$ V. This proportion is very small compared to the values

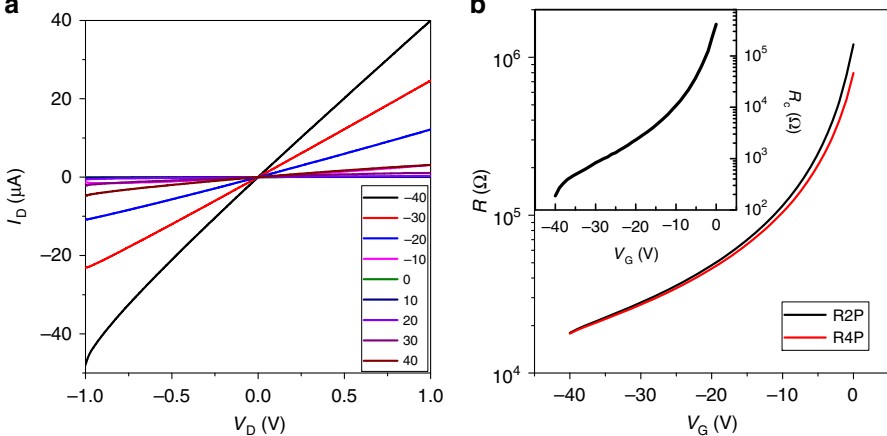

**Fig. 2** Contact resistance at Cr-BP interface. **a** Output curves of the BP device with increasing $V_G$ values with a step of 10 V. **b** Two-probe and four-probe device resistance to extract the contact resistance, inset shows the obtained contact resistance plot

obtained from TMDCs mainly due to the small band gap and the weak pinning factor of BP[22]. These results suggest that the contacts did not play a major role in the BP device, and the observed super-linear behavior was due to channel transport.

Another possible mechanism for the super-linear I–V behavior might be hopping transport in the BP channel which occurs due to the charge traps along the channel-dielectric interface[24]. In hopping transports, trapped charges are released with increasing temperature and electric field, leading to an increase in conductivity with bias. In fact, a 2D Mott's variable range hopping mechanism was recently reported as describing the carrier transport mechanism for $SiO_2$-supported thermoelectric BP devices[24,25]. However, the observed hopping transport occurs at low electric fields where the energy difference between the trap states and charge carriers is very large, while this difference becomes negligible at the large field values where energetic carriers can easily surmount the trap sites. We additionally observed a negligible hysteresis window for the BP devices (see the Supplementary Figure 1) because the BP-hBN interface was ultra-clean. These results supported the absence of hopping transports in our devices. Besides these, there might be possibility of other artifacts such as short channel effect in our BP devices. Miao et al. observed a weak short channel effect in the ultra-narrow channel (20 nm) BP devices[26]. Our measured shortest channel BP device is over two orders longer (over 2 μm) than observed short channel BP devices, and thereby, we rule out the possibility of short channel effect.

**Electrical field dependent magneto-transport measurements.** Unlike semiconducting TMDCs, multilayer BP possesses a very narrow band gap of 0.3 eV. Its high carrier mobility at room temperature could facilitate CM under a high electric field[9]. The contribution of CM to carrier transports was explored experimentally by measuring the carrier density in the BP device. We applied magneto-transport measurements to the hBN-encapsulated Hall-patterned BP devices, as shown in Fig. 3a, b. During the measurements, the magnetic field (**B**) was swept from −3 to 3 T at $V_G = -50$ V to measure the Hall voltage ($V_{xy}$) and channel voltage ($V_{xx}$) drop during the application of a fixed $I_D$ values from μA level to the breakdown point, as shown in Fig. 3c. The increasing $I_D$–$V_{xx}$ trend in the main panel of Fig. 3c indicates the super-linear behavior. The linearity of the plots at different $I_D$ values between $V_{xy}$ and **B**, as shown in the inset of Fig. 3c,

indicates an intimate interface between Cr-BP[27]. The Hall carrier density ($n_H$) was computed from the inset in Fig. 3c using the equation (2),

$$n_H = \left(d\mathbf{B}/dR_{xy}\right)/e \tag{2}$$

where $e$ is the electron charge and $R_{xy}$ is the Hall resistance of the BP device[25]. The $n_H$ values extracted at all applied $I_D$ points were assembled in Fig. 3d as a function of electric fields ($F = V_{xx}/L$), where $L$ is the channel length. Interestingly, we did not observe any changes in $n_H$ unless the applied field values were increased beyond 0.6 V μm$^{-1}$, at which point a sudden rise in $n_H$ occurred, as indicated by the shaded area. An obvious exponentially increasing trend was observed between $n_H$ and log $F$ at high fields, confirming carrier generation at a high field with an onset of ~0.6 V μm$^{-1}$. $n_H$ increased several-folds, as indicated by the CM factor (the ratio between the average carrier density at a low field ~2.5 × $10^{12}$ cm$^{-2}$ and the carrier density at a particular field) shown in Fig. 3d. The Hall carrier mobility ($\mu_H$) of this device was 250 cm$^2$ V$^{-1}$s$^{-1}$ at low-field region, that is extracted by the equation (3),

$$\mu_H = 1/(R_{xx} \times n_H \times e) \tag{3}$$

The reader might suspect that this increase in $n_H$ at a high field may have been due to the interface cleaning or contact improvement via self-heating[28]. This possibility was ruled out by measuring $n_H$ at a low field, then slowly increasing the field strength to high field values, then repeating the low-field measurement for comparison to the initial results. These results revealed very small deviation, merely 10% change in the $n_H$ values especially for $V_G$ higher than −6 V, (see the Supplementary Figure 8), confirming that the rise in $n_H$ was primarily due to impact ionization by hot carriers in BP devices at high electric fields.

The impact ionization mechanism can be further explored in view of the energy band diagram of BP, as shown in Fig. 3e. The large electric field applied to the device accelerates the charge carriers (path 1), which gain kinetic energy with partial loss to phonons (mainly optical phonons). The energetic hot carriers scatter off the lattice (path 2), and excite the electron-hole pairs (path 3) through the impact scattering mechanism, thereby increasing $n_H$ in the channel. Lastly, the increased majority and minority charge carriers are collected at the respective electrodes

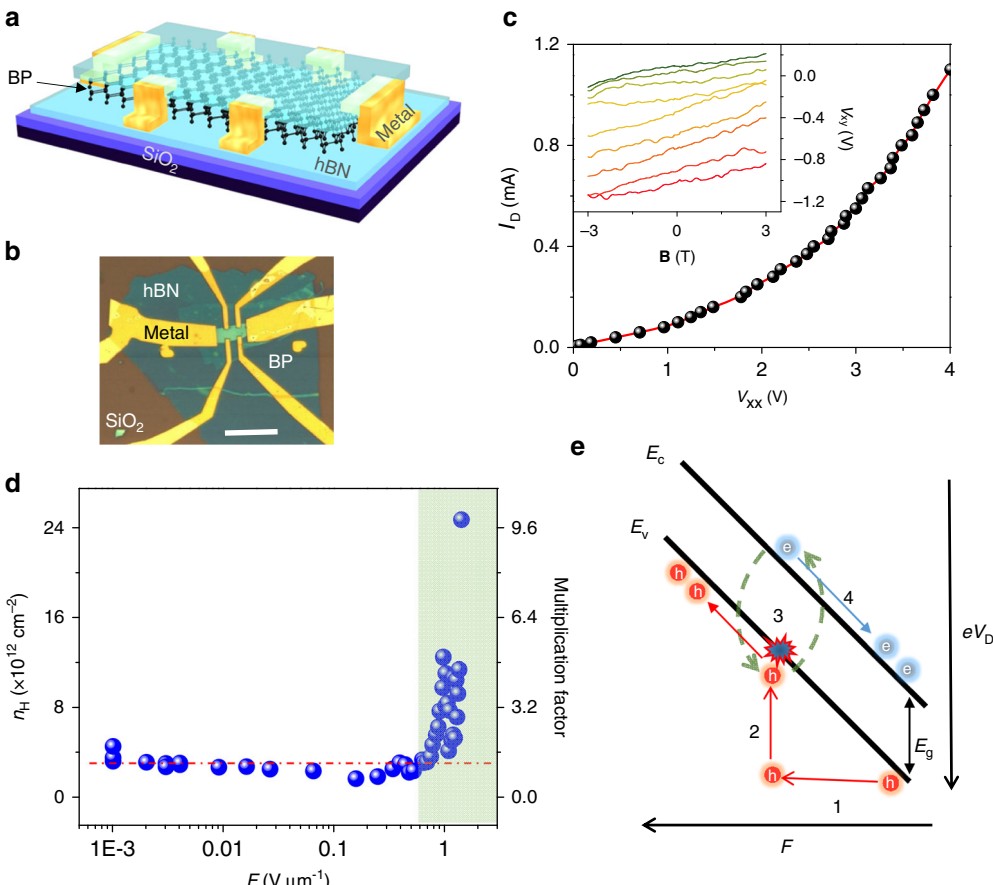

**Fig. 3** Magneto-transport measurements collected from a multilayer BP device. **a**, **b** show schematic and optical images, respectively, of hBN–encapsulated Hall-patterned BP device. The thickness of hBN top, BP, and hBN bottom is 9, 28, 15 nm, respectively, and scale bar is 10 μm. **c** The channel voltage drop ($V_{xx}$) and the applied source-drain current ($I_D$). Inset: Hall voltage ($V_{xy}$) as a function of the magnetic field at $V_G = -50$ V at a given $I_D$. For ease of viewing, a few data plots are shown from 0.2 mA (green) to 1 mA (red) $I_D$ with step of 0.1 mA. **d** Carrier concentration ($n_H$) as a function of the applied $I_D$ and electric field, where blue circles are data points and red dashed line indicates mean $n_H$ value at low field. **e** Energy Band diagram showing carrier multiplication in the BP channel, where $E_c$, $E_v$, and $E_g$ represent conduction band edge, valance band edge and energy band gap of BP, respectively

(path 4). The excess kinetic energy between electrons and holes is distributed based on their effective masses, such that the lighter carriers receive more of the excess energy[7]. In the case of BP, the effective mass of the holes ($m_x = 0.076m_0$ and $m_y = 0.648m_0$) is smaller than that of the electrons ($m_x = 0.0826m_0$ and $m_y = 1.027m_0$) along both of the in-plane directions[29]. Therefore, impact ionization by energetic hot holes in the multilayer BP channel is highly probable due to its p-type polarity. Furthermore, impact ionization is realized when the rate of impact ionization exceeds the cooling rate and other relaxation processes associated with the hot carriers in a device[7,8]. In the hBN-encapsulated BP devices examined here, the confined geometry of layered BP channel together with the ultraflat surface and high optical phonon energy of hBN (150–200 meV) compared to the corresponding energy of SiO₂ (60–80 meV) inhibits hot carrier relaxation, such that the rate of impact ionization becomes competitive with the rate of carrier cooling. In addition to this, impact ionization is desirable in low dielectric screening and low carrier density conditions, since they give rise to strong electron–electron interactions in a channel. Impact ionization is, therefore, expected to be more prominent in a multilayer BP channel supported on relatively thick and low-k dielectric materials such as hBN (3.5) and/or SiO₂ (3.9) compared to those of high-k dielectrics like Al₂O₃ and HfO₂ etc. This explains the observation of super-linear I–V characteristics instead of current saturation in our hBN encapsulated multilayer BP devices supported on 285 nm thick SiO₂ (also see S1).

## Discussion

Ionization across the band gap of the material can be achieved provided that the hot carriers overcame a certain energy threshold. The threshold of the energy ($E_{TH}$) is estimated using the equation (4),

$$E_{TH} \approx 1.5E_g/(q\lambda_{OP}) \qquad (4)$$

where $E_g$ is the band gap, $q$ is the elementary charge and $\lambda_{OP}$ is the mean free path (MFP) of electron with respect to the optical phonon scattering in the material[30]. Using a 0.3 eV band gap for multilayer BP and a $\lambda_{OP}$ in the range of 66–83 nm, depending on directional anisotropy of BP[31], a lower bound $E_{TH}$ of 7–5.5 V μm⁻¹ was computed. Note that $\lambda_{OP}$ decreased with increasing operating temperature[31]. The extracted $E_{TH}$ range is very close and, in some cases, well below the breakdown field of the measured BP devices (see Fig. 1c, d), confirming that impact ionization can become the predominant transport mechanism at high electrical fields to observe super-linear I–V trend in multilayer BP devices. The onset field of impact ionization in Hall-bar devices, as shown in Fig. 3d, was smaller than the threshold value. We think the onset field value in Fig. 3d was underestimated, as it did not include the voltage drop along the contacts, and a real onset for CM might be closer to the energy threshold. The ionization threshold field values for semiconducting TMDCs such as MoS₂, WS₂, MoSe₂, and WSe₂ were typically very high due to the wide band gap (1–2 eV) and

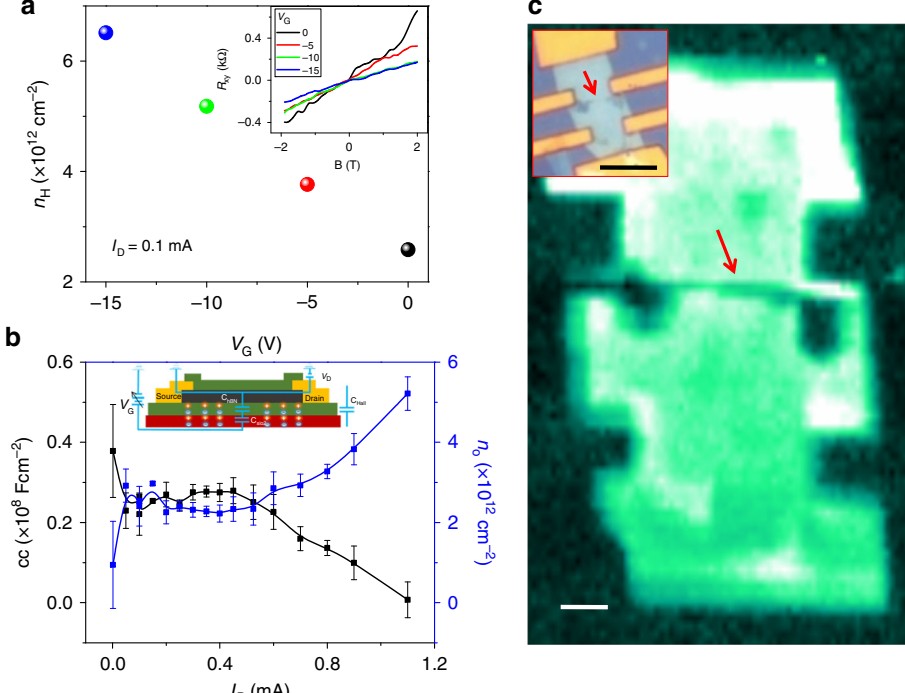

**Fig. 4** Capacitive coupling and breakdown of a BP channel. **a** Carrier density modulation by $V_G$ at $I_D = 0.1$ mA. Inset: $R_{xy}$ as a function of the applied magnetic field at a given $V_G$. **b** The extracted capacitive coupling (CC) and residual doping density ($n_o$) of a BP channel under the applied $I_D$, where the inset shows back gate capacitive effect on an $h$BN-sandwiched BP device. Error bars are obtained from four replicated data sets. **c** False–color Raman intensity mapping of a broken BP device, scale bar is 2 μm, where the red arrow indicates a crack along the channel, and the inset shows an optical micrograph of the same device after Joule breakdown with a scale bar of 5 μm

very low optical phonon MFP for these materials. For monolayer $MoS_2$, the ionization threshold field was 200 V μm$^{-1}$, very high compared to that of multilayer BP[30]. It is, therefore, difficult to realize impact ionization in semiconducting TMDCs. Very recently, Barati et al. reported CM by hot carriers in a small band-offset (1 eV) $WSe_2/MoSe_2$ heterojunction[4]. The nature of band alignment in the $WSe_2/MoSe_2$ heterostructures was such that the CM could be only observed in a reverse bias regime. The BP devices studied here provided an advantage of simple device fabrication, could be operated along both forward and reverse bias conditions, and might exhibit a higher conversion efficiency in photocells due to the direct band gap of BP, when compared to $WSe_2/MoSe_2$ heterostructures.

Next, we studied the gating of the CM by measuring $n_H$ for $h$BN-encapsulated BP devices at four different $V_G$ values: 0, −5, −10, and −20 V as shown in the inset of Fig. 4a at a fixed $I_{SD}$ value at which $n_H$ increased linearly with $V_G$. The CC between the channel and the back gate dielectric was extracted by using the equation (5),

$$n_H = n_o + C_H \times V_G / e \qquad (5)$$

where $n_0$ is the residual doping density of the BP channel, and $C_H$ is the Hall capacitance that denotes the CC between the BP channel and the gate dielectric[32]. The linear slope and intercept were used to extract CC and $n_0$ as a function of $I_D$. At low current values, i.e., 0.1 mA $I_D$, the CC and $n_0$ values were $0.28 \times 10^{-8}$ F cm$^{-2}$ and $2.83 \times 10^{12}$ cm$^{-2}$, respectively. The CC value fell in the range of capacitances typically obtained from the parallel plate model and was very close to the value reported for monolayer $MoS_2$[32]. Similarly, we computed the CC and $n_0$ at higher applied current values as shown in Fig. 4b. The CC ($n_0$) values first did not show notable changes at lower applied $I_D$; however, a sudden

drop (rise) was observed as the $I_D$ value increased. The rather flat CC trend at low $I_D$ values was attributed to the constant number of charge carriers in the BP channel, indicating a strong coupling between BP and $h$BN; however, as $I_D$ increased, the dielectric lost a control over the channel mainly generating more carriers by impact ionization in the BP channel, thereby decreasing CC. Similarly, the increase in $n_0$ at large $I_D$ confirmed CM via impact ionization by hot carriers in the BP channel.

Finally, we examined the broken device by optical microscopy and Raman spectroscopy with a spatial focus on the position at which BP experienced damage, indicating the formation of a hot-spot. A Raman intensity mapping of the $h$BN-encapsulated BP device was obtained, revealing an obvious crack in the BP channel along the inner contacts of the Hall pattern, as shown in Fig. 4c. The two-probe $SiO_2$ supported BP devices suffered similar crack near the biased contact (see the Supplementary Figure 9). These cracks were attributed to the current crowding[33], non-homogeneous thermal spreading[17,34], and the formation of an abrupt p-n junction near the biased contacts[35,36]. In this case, cracks may have been formed due to uneven doping of the BP channel induced by impact ionization by hot carriers. After electron-hole pair excitation, the electrons and holes drifted towards the drain and source contacts, respectively, during which the proximal region of the drain contact becomes more resistive, leading to the formation of hotspots. The CM and heat spreading inside the channel could be further explored in future studies by incorporating spatial techniques.

In summary, we realized an unusual CM in multilayer BP devices under high field strengths. Magneto-transport measurements were used to confirm the presence of CM due to impact ionization by highly accelerated charge carriers thanks to the narrow (0.3 eV) and direct band gap of multilayer BP. Lastly, CM affected the spatial thermal spreading in the BP channel and its

capacitive coupling to the dielectric. This study will be highly significant to realize highly efficient optoelectronic devices like photovoltaic cell, photodetectors and solar cells based on layered materials.

## Methods

**Device fabrication.** The fabrication of the $h$BN/BP/$h$BN began with mechanical exfoliation of multilayer $h$BN flakes on p-type Si substrate coated with a 285 nm thick thermal oxide layer of SiO$_2$. Multilayer BP flakes were exfoliated on poly-dimethylsiloxane (PDMS) film, over which, only rectangle or square BP flakes were selected to avoid current spreading during electrical measurements. The desired BP flakes were transferred over $h$BN by PDMS stamping technique. Electrodes were patterned in desired geometry over BP flakes by electron-beam lithography, after that contact metals were deposited by electron beam deposition. Finally, the top $h$BN was transferred in similar fashion. For further analysis about the effectiveness of $h$BN capping, see the Supplementary Figures 10 and 11. The exfoliation and transfer of BP was carried out in controlled environments glove box to preserve BP from external perturbations.

**Data availability.** The data sets generated and/or analysed during the current study are available from the corresponding author on request.

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

## Acknowledgements

This work was supported by the Global Research Laboratory (GRL) Program (2016K1A1A2912707) and Global Frontier R&D Program (2013M3A6B1078873), both funded by the Ministry of Science, ICT & Future Planning via National Research Foundation of Korea (NRF).

## Author contributions

F. A., Y. D. K., J. H., and W. J. Y. conceived of the research project, E. H., J. H., and W. J. Y. supervised the experiment, and wrote the manuscript. F. A., and Z. Y., performed the device fabrication. F. A., Z. Y., and P. H. performed the electrical and magneto-transport characterization under the supervision of H. Y., and W. J. Y.

## Additional information

**Competing interests:** The authors declare no competing interests.

