## [Peer Review File · Nature Communications]

Reviewers' comments:

Reviewer #1 (Remarks to the Author):

The authors claim that carrier multiplication (CM) process by impact ionization in thin-film field-effect transistor (FET) of black phosphorus (BP) are observed in the high electric field for the first time. Other possible carrier transport mechanisms such as contact resistance and variable hopping transport are discussed in high electric strength. The experimental magneto-transport investigation by Hall measurement and capacitive coupling measurement between the device channel and the back gate were performed with hBN (hexagonal Boron Nitride)-encapsulated BP devices to support the authors' claims.

Even though the reported topics are interesting to material scientists and solid-state physicists, I think that this paper is immature to be published in Nature Communication (reject), a high impact journal, due to following flaws of the authors' logic, and lack of novelty, and no demonstration regarding applications with the discovered phenomena.

- The authors claim that the experimental CM in a two-dimensional (2D) BP FET is unusually observed. However, to obtain the CM in the BP system, the thickness should be larger than six layers, allowed to a bulk for ~ 0.3 eV bandgap. Therefore, the authors cannot use the term, '2D' for CM.
- The range of the high electric field by a gate is ambiguous. The contact resistance of BT FETs seems high for $V_G = -20V$ to $0V$ from Fig. 2(b). Furthermore, the $nH - V_G$ results in Fig. 4(a) are measured from $V_G = -16V$ to $0V$ so that the effect of contact resistance is dubious. Therefore, the definition of high electric field is unclear.
- What are the specific values of V_G of various color lines in Fig. 2(a)? Please clarify.
- Please graph Fig. 2(b) with the same scale of y-axis including R_c for clear comparison to readers.
- The authors said in line 183 - 184, "These results revealed no significant differences between the nH values, as shown in Supplementary Fig. S4". However, in the cases of large V_G (larger than $-8V$) shows large deviation between 'first attempt' and 're-measured', more than 10% in my eye.
- There is no demonstration (even prototype demonstration) of interesting application. The last sentence in the manuscript, "This study will be highly critical for the realization of practical and durable future devices based on 2D materials." is very ambiguous and is not true.
- There is no provided information about the device thickness which is used for the measurement. The thickness range of the fabricated BP FET should be mentioned for being reproducible.
- Typos for data numbers
- * Line 170: Fig. 4(d) => Fig. 3(d)
- * Line 178: Fig. 4(d) => Fig. 3(d)
- * Line 235: 2.83×10^{-12} => should be 2.83×10^{12}

Reviewer #2 (Remarks to the Author):

The paper by Ahmed et al. reports on super-linear I-V characteristics in multilayer black phosphorus (BP) FETs. The backgated devices are p-type and consist of a stack of hBN/BP/hBN deposited on a 285 nm oxide (SiO_2) on doped silicon. A super-linear current is observed when applying a large field to the drain and this increases is seen until the irreversible breakdown of the channel. The authors performed magneto-transport measurements at $V_g = -50V$ and extracted the carrier density as a function of V_d . They observed constant density in the super-linear I-V regime and a sudden increase of density when the field reach $\sim 0.6V/\mu m$. The absence of a current saturation, the super-linear I-V and the sudden increase of charge carrier density are rationalized using a model based on impact ionization.

Assuming that charge multiplication (CM) is a big deal here for 2D-material FETs, the authors should at least mention that CM is known for semiconductors and expected when low screening

and doping conditions are met. Clearly the device fabricated here did not select device dimensions in favor of low screening (i.e. oxide thickness is ~ 300 nm and gate screening is therefore strong) and therefore the CM conclusion is surprising. To reach the CM conclusion, the authors explored all kind of options, such as hopping and contact issues, which help rule out unlikely hypotheses about the super-linear I-V and this is good. However, all the experiments on the p-type BP FETs use the wrong direction for the drain bias! This mistake alone might explain the super-linear behavior observed. That is, a positive bias to the drain side of a p-type FET makes the field poorly defined in the region of the drain. As described in most textbooks on FETs (e.g, Sze), the field on the drain increases with positive drain bias and this brings issues such as super-linear I-V because the carrier density increases on the drain side with applied V_d . The expected saturation for p-type FETs is seen when the bias is negative (as it should) and this is due to a pinch-off because the negative drain voltage deplete carriers in the channel. Of course, this saturation behavior is for long channel situation. For short channel, the behavior shows weak or no saturation and sometimes even super-linear I-V. The experiments presented here and the discussion are poor and cannot be used to support a CM mechanism. I do not recommend publication of this work in any form unless more evidence are given.

Major Points:

- 1) In addition of using proper biasing conditions for the measurements of p-type FET, the authors should explore also the effect of screening and carrier density on the I-Vs. I am surprised that only one device geometry is used here, which makes the study quite shallow as far as CM is concerned. As explained above, the observation of super-linear I-V is not sufficient to conclude about CM and more work is required to understand the physics of these devices.
- 2) The thickness of the layers, both hBN and BP should be given. These are important parameters to determine before one can understand and reproduce the electrostatics of the devices.

Minor points:

- 1) On p. 5 is described the Raman spectra. The term "atomic oscillations" to describe BP vibrations is poorly chosen and probably wrong.
- 2) Some references are wrong or poorly selected (e.g. environmental sensitivity ascribed to Ref. 16 and 17), which makes me wonder if the authors have a good knowledge of the literature on both BP and FETs.

Reviewer #3 (Remarks to the Author):

In this manuscript, the authors studied the impact ionization in black phosphorus. The data are coherent and support the conclusions well. For these reasons, I suggest a minor revision of the paper before it can be accepted by Nature Communications. Following are the questions that need to be addressed:

1. On Page 6, the authors mention that "an alternative carrier transport mechanism is present and capable of masking current saturation in BP." Can the authors explain what is the carrier transport mechanism?
2. On Page 9, the authors mention "the effective mass of the holes is smaller than that of the electrons". Can the authors indicate the effective mass of electrons and holes in BP?
3. What is the thickness of BP in the device used in the paper? More material characterization is needed.

Reviewers' comments:

Reviewer #1 (Remarks to the Author):

The authors claim that carrier multiplication (CM) process by impact ionization in thin-film field-effect transistor (FET) of black phosphorus (BP) are observed in the high electric field for the first time. Other possible carrier transport mechanisms such as contact resistance and variable hopping transport are discussed in high electric strength. The experimental magneto-transport investigation by Hall measurement and capacitive coupling measurement between the device channel and the back gate were performed with hBN (hexagonal Boron Nitride)-encapsulated BP devices to support the authors' claims.

Even though the reported topics are interesting to material scientists and solid-state physicists, I think that this paper is immature to be published in Nature Communication (reject), a high impact journal, due to following flaws of the authors' logic, and lack of novelty, and no demonstration regarding applications with the discovered phenomena.

Reply:

We thank the reviewer for a careful reading of our manuscript and appreciate acknowledging the scientific impact of our work to material scientists and solid-state physicists. Much more than that, we truly appreciate several very important comments and criticisms from the reviewer that we overlooked in our original manuscript. Faithfully following all the comments and suggestions from the reviewer, **we have performed additional experiments, addressed all the issues, and significantly revised the manuscript with several new discussions in the main manuscript as well as in the supplemental materials.** We believe that the revised manuscript, fully addressing the specific concerns of the reviewer, better showcase the possible technological impact. We have tried to best show the novelty and technological applicability of our work. We value the comments of the reviewer on how we present our work better.

The primary focus of current study is to identify and experimentally prove the carrier multiplication (CM) phenomenon in intrinsic BP devices. CM is highly important for solid-state physics and for realizing high performance opto-electronics devices. To the best of our knowledge, such an important phenomenon has not been reported previously in semiconducting 2D materials in their pristine form. Previously, number of research groups have reported the intrinsic carrier transport such as Shubnikov-de Hass (SdH) oscillations, quantum Hall effect, fractional quantum Hall effect and so on [Li *et al.*, Nat. Nanotech., 10, 608-613, 2015 & Li *et al.* Nat. Nanotech., 11, 593-597, 2016 & Chen *et al.*, Nat. Comm., 6, 7315, 2015 & Yang *et al.*, Nano Lett., 18, 299-234, 2018] in hBN encapsulated BP devices. All the mentioned studies are carried out at low temperature and low bias in the presence of magnetic field and these studies are helpful to understand the device physics. However, we report for the first time intrinsic transport phenomenon in BP at practical operating conditions i.e. high electric field, where we not only identify the super-linear I-Vs, but also experimentally verify the responsible mechanism by magneto-transport measurements. Even though, the super-linear I-Vs at high field in BP was also observed by other groups [experimentally by Engel *et al.*, Nano Lett. 15, 6785–6788 (2015) and theoretically by Trushkov *et al.*, Phys. Rev. B 95, 75436 (2017)], but the mechanism behind it was never elaborated before. Based on these facts, we understand that our study is practical as well as novel in 2D research field, and has a wide scope in optical applications.

Q: The authors claim that the experimental CM in a two-dimensional (2D) BP FET is unusually observed. However, to obtain the CM in the BP system, the thickness should be larger than six layers, allowed to a bulk for ~ 0.3 eV bandgap. Therefore, the authors cannot use the term, '2D' for CM.

Reply:

We thank the reviewer for comment. Normally, 2D term is used for substances with one dimension in the range of a few nanometers or less. Therefore, we replace the term '2D' with 'layered' and 'multilayer' wherever it suits.

Q: The range of the high electric field by a gate is ambiguous. The contact resistance of BP FETs seems high for $V_G = -20V$ to $0V$ from Fig. 2(b). Furthermore, the $nH - V_G$ results in Fig. 4(a) are measured from $V_G = -16V$ to $0V$ so that the effect of contact resistance is dubious. Therefore, the definition of high electric field is unclear.

Reply:

We thank the reviewer for the comment. It is a matter of fact that contact resistance (R_c) is a bottleneck in pursuit of high performance devices and to realize intrinsic transport in layered materials. This effect is very severe in wide band gap layered materials like Transition metal dichalcogenides (MoS_2 , WSe_2 , $MoTe_2$ and so on) due to presence of very high Schottky barrier and strong pinning effect along channel-electrode contacts. On the other end, the BP based devices exhibit very small R_c value with most of the commonly used metals mainly due to smaller band gap and weak pinning effect. For example, the R_c for Cr-BP contact at $V_G = -20V$ is < 10 k Ω as shown in inset of Fig. 2(b), that value is several times smaller as compared to the R_c of Cr- MoS_2 contacts (Ahmed *et al.*, *Nanoscale*, **7**, 9222-9228, 2015). Therefore, we think the effect of R_c on BP based devices is very small even though the applied V_G values are high such as $-20V$ to $0V$ compared to other 2D semiconducting materials.

In addition to these, the data set in Fig. 4(a) is assembled by Hall-bar measurements, where voltage drop (V_{xx}) across the inner electrodes is computed by applying constant current across outer electrodes, clearly indicating that it does not include the effect of voltage drop along the contacts. Therefore Fig. 4(a) plot does not include the effect of R_c anymore. It is imperative to note here that, the plot shown in Fig. 4(a) is at low applied field conditions i.e. $100 \mu A$, that is one of the first few initial measurement points (low field) as shown in Fig. 4(b). It does not depict the high field data set, as explained in its corresponding text.

Based on these arguments, it is safer to say that the role of R_c in our study is very minute.

Q: What are the specific values of V_G of various color lines in Fig. 2(a)? Please clarify.

Reply:

We thank the reviewer for the comment. The V_G values from 40 to $-40V$ is applied to compile output curves, as shown in Fig. 2(a). We have provided specific V_G values with all the colors.

Q: Please graph Fig. 2(b) with the same scale of y-axis including R_c for clear comparison to readers.

Reply:

We thank the reviewer for comment. We have modified the Fig. 2(b) with same y-axis scale, and divided into two figures; Fig. 2(b) shows 2 probe and 4 probe resistance and inset shows R_c plot.

Q: The authors said in line 183 - 184, “These results revealed no significant differences between the n_H values, as shown in Supplementary Fig. S4”. However, in the cases of large V_G (larger than $-8V$) shows large deviation between ‘first attempt’ and ‘re-measured’, more than 10% in my eye.

Reply:

We thank the reviewer for the comment. We agree that the S4 (which is now S9 in revised draft) shows approximately 10~12% deviation especially for $V_G > -6V$, perhaps due to small carrier density and slightly higher R_c at high V_G . Nonetheless, the 10~12% is very small change as compared to increase in carrier density at high field as shown in Fig. 3(d). For more clarification, we revised the above mentioned line as “**These results revealed very small deviation, merely 10% change in the n_H values especially for $V_G > -6V$, as shown in supplementary Fig. S9**”. In addition to this, we also included one additional plot in supporting information S9 that shows change in n_H , also shown here;

FIG. S9 (a) Hall carrier density measured initially and after the effect of Joule heating at $I_D = 0.1$ mA. (b) Change in carrier density at different V_G .

Q: There is no demonstration (even prototype demonstration) of interesting application. The last sentence in the manuscript, “This study will be highly critical for the realization of practical and durable future devices based on 2D materials.” is very ambiguous and is not true.

Reply:

We thank the reviewer for the comment. It is a matter of fact that, carrier multiplication is highly useful for opto-electronic devices like photovoltaic cell, photodetectors and solar cells. To the best of our knowledge, the CM phenomenon has rarely been observed in semiconducting 2D materials in their pristine form. The primary purpose for the current article is to observe and experimentally prove CM phenomenon in BP devices that we tried to prove by measuring field dependent magneto-transport measurements. We understand that the application of such an interesting phenomenon does not follow in the domain of current study, as it requires an insightful and deep explanation of opto-electronic results. In fact, we are currently preparing another manuscript that is mainly based on in-depth study of application part of CM. For reviewer’s assurance we are providing the quick results of photo-response as a function of applied field values, where we can observe a clear enhancement in photocurrent with field due to

CM in BP channel. See the following figures.

Fig. (a) OM image of a multilayer BP (20 nm thick) device, (b) Photocurrent (PC) as a function of increasing field at $V_G = 0$. Note that, photocurrent (PC) is the difference between light and dark current $PC = I_{\text{light}} - I_{\text{dark}}$, and for this we used 550 nm laser illumination.

Moreover, in the above mentioned line, the terms ‘practical and durable’ refer to the practical operating conditions *i.e.* higher applied electrical field to the BP device. To remove any doubt, we revised the above sentence with ‘**This study will be highly significant to realize highly efficient opto-electronic devices like photovoltaic cell, photodetectors and solar cells based on layered materials.**’

Q: There is no provided information about the device thickness which is used for the measurement. The thickness range of the fabricated BP FET should be mentioned for being reproducible.

Reply:

We thank the reviewer for the comment. The detailed thickness information of our representative BP devices, whose data sets are shown in manuscript, is given in supporting information S4. Where we have compiled a table to enlist the thickness information of measured BP devices, as;

Device #	Geometry	Thickness (nm)			Remarks
		Bottom h BN	BP	Top h BN	
1	h BN/BP/ h BN	26	24	20	Two-probe device; Data set shown in Fig. 1 (b) and (c) for negative V_D breakdown
2	h BN/BP/ h BN	30	36	25	Two-probe device; Data set shown in Fig. 1 (d) and S5 for positive V_D breakdown.
3	h BN/BP/ h BN	15	28	9	Hall bar device; Data set shown in Fig. 3(c) and (d)
4	h BN/BP/ h BN	40	30	18	Hall bar device; Data set shown in Fig. 4 (a) and (b)
5	h BN/BP	25	6	-	Two terminal device; Data set shown in S7 for different dielectric comparison.
6	SiO ₂ /BP	285	33	-	Two terminal devices; Data set shown in S8 for anisotropic breakdown comparison.

As an example, the optical microscopic (OM) image and AFM images of two terminal *h*BN encapsulated BP device (device #1) is shown in following figures. The height profile of individual flake scanned by AFM is shown in given plots, wherein, *h*BN top, BP channel and *h*BN bottom are 20 nm, 24 nm and 26 nm thick, as indicated by black, red and blue colors respectively.

FIG. S4 (a) and (b) are OM and AFM images of $h\text{BN}$ encapsulated BP device.

We have provided the thickness information of all the measured devices in the revised manuscript on page 6, line 7 from bottom, as;

“We repeated the experiments to the SiO_2 and/or $h\text{BN}$ supported back gated BP devices of more than 30 different thicknesses (6 ~ 42 nm thick BP flakes)”

In captions of Fig. 3(b) of manuscript as;

“The thickness of $h\text{BN}$ top, BP and $h\text{BN}$ bottom is 9, 28, 15 nm respectively.”

Q: Typos for data numbers

* Line 170: Fig. 4(d) => Fig. 3(d)

* Line 178: Fig. 4(d) => Fig. 3(d)

* Line 235: 2.83×10^{-12} => should be 2.83×10^{12}

Ans: We thank the reviewer for identifying the typos in manuscript.

Reviewer #2 (Remarks to the Author):

The paper by Ahmed et al. reports on super-linear I-V characteristics in multilayer black

phosphorus (BP) FETs. The backgated devices are p-type and consist of a stack of hBN/BP/hBN deposited on a 285 nm oxide (SiO₂) on doped silicon. A super-linear current is observed when applying a large field to the drain and this increases is seen until the irreversible breakdown of the channel. The authors performed magneto-transport measurements at $V_g = -50V$ and extracted the carrier density as a function of V_d . They observed constant density in the super-linear I-V regime and a sudden increase of density when the field reach $\sim 0.6V/\mu m$. The absence of a current saturation, the super-linear I-V and the sudden increase of charge carrier density are rationalized using a model based on impact ionization.

Assuming that charge multiplication (CM) is a big deal here for 2D-material FETs, the authors should at least mention that CM is known for semiconductors and expected when low screening and doping conditions are met. Clearly the device fabricated here did not select device dimensions in favor of low screening (i.e. oxide thickness is ~ 300 nm and gate screening is therefore strong) and therefore the CM conclusion is surprising. To reach the CM conclusion, the authors explored all kind of options, such as hopping and contact issues, which help rule out unlikely hypotheses about the super-linear I-V and this is good. However, all the experiments on the p-type BP FETs use the wrong direction for the drain bias! This mistake alone might explain the super-linear behavior observed. That is, a positive bias to the drain side of a p-type FET makes the field poorly defined in the region of the drain. As described in most textbooks on FETs (e.g, Sze), the field on the drain increases with positive drain bias and this brings issues such as super-linear I-V because the carrier density increases on the drain side with applied V_d . The expected saturation for p-type FETs is seen when the bias is negative (as it should) and this is due to a pinch-off because the negative drain voltage deplete carriers in the channel. Of course, this saturation behavior is for long channel situation. For short channel, the behavior shows weak or no saturation and sometimes even super-linear I-V. The experiments presented here and the discussion are poor and cannot be used to support a CM mechanism. I do not recommend publication of this work in any form unless more evidence are given.

Reply:

We thank the review for the helpful and detailed comments. The reviewer has rightly pointed out number of fundamental points here.

We agree with the reviewer's point of the optimal condition for CM, i.e., low screening and low doping. Our samples are not intentionally doped and their carrier densities are tuned by the gate voltage. As expected, we have observed the super-linear behavior at low gate voltages (i.e., low doping cases). Regarding the screening effects, we believe that our samples are really in lower screening regime compared to the typical semiconductors observed CM. Since the samples are sandwiched by SiO₂ and air (neglecting the screening effect by thin hBN) the average dielectric constant is about 2.5 which is much smaller than those of the semiconductors (usually the dielectric constants of the semiconductors are ~ 10). Thus, we expect the stronger electron-electron interaction in our samples, which gives rise to the effective carrier multiplication in BP.

Firstly, we added the following lines in manuscript at lines 11~13 of introduction as;
“The efficient CM can be expected in a semiconducting device, when a low dielectric screening and low doping conditions are realized that results in strong electron-electron interactions.”
And on page #10, lines 7~14.

“In addition to this, impact ionization is desirable in a low dielectric screening and low carrier density conditions, since they give rise to strong electron-electron interactions in a channel. Impact ionization is, therefore, expected to be more prominent in a multilayer BP channel supported on relatively thick and low- k dielectric materials such as hBN (3.5) and/or SiO_2 (3.9) compared to those of high- k dielectrics like Al_2O_3 and HfO_2 etc. This explains the observation of super-linear I-V characteristics instead of current saturation in our hBN encapsulated multilayer BP devices supported on 285 nm thick SiO_2 .”

As long as choice of bias range is concerned, we agree with the reviewer that the negative bias region should be chosen to observe current saturation behavior in p-type material like BP. In fact, there are already number of publications, wherein, the current saturation in BP FET especially in negative bias direction. We had already measured negative bias ($-V_D$) region to the number of BP devices for high field measurements, and yet there was no any sign of current saturation. By keeping in mind the p-type dominated nature of multilayer BP devices, we replaced the Fig. 1 (b) and (c) in revised manuscript with the data set measured towards the negative V_D sweep direction in device #1, where we can clearly observe super-linear I-V up to the breakdown point. In addition to these, for the readers’ clarification, we also provide the high field plot of positive V_D region in Fig. 1(d) (obtained from device #2). It also showed super-linear I-V. Based on these, we can say that the super-linear I-Vs can be obtained in BP devices in both forward and reverse bias regions provided low screening and doping conditions are met.

Fig. 1 Characterization of a multilayer BP device and its breakdown. (a) Schematic diagram illustrating the fabrication steps of a back-gated *h*BN-encapsulated BP FET device. (b) 2D contour plot representing the device current (I_D) as a function of various V_D (from 0 to -1 V) and V_G (from 40 to -40 V), where *n* and *p* indicate electron and hole branches and their corresponding band alignment positions of the BP channel at given bias conditions. (c) High electric field Joule breakdown at $V_G = 0$ towards negative field region in the BP device (#1) (d) Joule breakdown towards positive field region in BP device (#2).

Besides this, we moved the low field data set of device #2 to the supporting information S5.

FIG. S5 (a) OM image of *h*BN encapsulated BP device (#2), where the black rectangle indicates measured device. (b) Low electrical field 2D contour plot of BP device. (c) Band position of BP at specific bias condition.

We think, the above mentioned case explained from reviewer might hold for diodes, bulk devices and Schottky barrier devices. For diodes like p-n junction, where applying negative bias will generate carrier multiplication prior to avalanche breakdown. In case of bulk devices like especially short channel Si body FETs, where contacts are imbedded by opposite polarity and back-to-back junctions are formed along body-contact interface (*S. M. Sze, Physics of semiconductor devices, second edition and B. G. Streetman and S. K. Banerjee, Solid state electronic devices, sixth edition*). There might be possibility of carrier generation along the interface by applying large reverse bias. Lastly, the Schottky barrier devices may also induce CM due to large R_c as explained in manuscript.

Since BP FETs measured here are thin compared to bulk devices and does not contain implanted electrodes like Si body FETs. Besides this, the BP devices exhibit very small R_c value due to Ohmic-like interfacial barriers, as indicated by linear I_D - V_D plot specially at low field as shown in Fig. 2(a) and measured R_c value in Fig. 2(b). Therefore, we think there is no drastic difference in BP devices by switching polarity especially at low field. However, at high field the output curves deviate towards non-linearity at positive and negative bias conditions due to significant band bending and thermal effects as shown in supporting Fig. S6. Based on our obtained measurement results, we think it is elusive to observe current saturation in pristine BP FET in positive as well as negative bias region until and unless strong screening conditions such

as high- k and very thin gate dielectric are realized. But, it will bring fabrication complexities and gate leakage problems, also explained in supporting information S1.

Major Points:

1) In addition of using proper biasing conditions for the measurements of p-type FET, the authors should explore also the effect of screening and carrier density on the I-Vs. I am surprised that only one device geometry is used here, which makes the study quite shallow as far as CM is concerned. As explained above, the observation of super-linear I-V is not sufficient to conclude about CM and more work is required to understand the physics of these devices.

Reply:

We thank the reviewer for the comment. The screening and carrier density are highly critical parameters for observing super-linear I-V plots. We have carried out high field transport to the BP devices from the various viewpoints of device physics such as; dielectric engineering (affects charge screening), different gating conditions (carrier density) and in-plane anisotropy of BP.

Previously there have been number of publications wherein, the current saturation in BP devices was reported, as summarized in given table. However, in most of these studies, the very thin and/or high- k dielectric and narrow BP channel are used to realize very strong charge screening effect. These special kind of devices might be preferable for radio-frequency applications but in the meantime, brings fabrication complexities and gate leakage problems.

#	Reference	BP channel		Gate dielectric		
		Gate length (μm)	Thickness (nm)	Material	Dielectric constant	Thickness (nm)
1	L. Li et al., Nat. Nanotechnol. 2014, 9, 372–377	4.5	5	SiO ₂	3.9	90
2	W. Zhu et al., Nano Lett. 2015, 15, 1883–1890	2.7	15	Al ₂ O ₃	10	25
3	H. Wang et al., Nano Lett 2014, 14, 6424–6429	0.3	8.5	HfO ₂	16.5	21
4	S. Das et al., ACS Nano 2014, 11730–11738	2	1.9	SiO ₂	3.9	20
5	W. Zhu et al., Nano Lett. 2016, 16, 2301–2306	0.5	13	Al ₂ O ₃	10	25

However, the BP devices fabricated in our study are supported by $h\text{BN}$ and/or SiO₂ dielectric, both of which are low- k dielectric materials (3.5 and 3.9 respectively), therefore, the screening effect is weak. As the reviewer rightly pointed that the CM phenomenon is preferable in low screening conditions. And this alone explains the observation of super-linear I-V trends instead of current saturation in our BP devices.

Moreover, the carrier density is another critical parameter for CM phenomenon. We also agree with the reviewer that the low carrier density is preferable for CM, since it results in strong electron-electron interactions. In fact, we applied different gating conditions (0 to -50 V_G) to induce more carrier density in BP channel at high electrical field, wherein, we did not observe any sign of current saturation, as shown in S6, as;

FIG. S6 (a) and (b) are high field output plots for positive and negative V_D regions respectively at given V_G .

In addition to this, we computed gate induced carrier density $\{n_G = C \times (V_G - V_{Th})\}$, where V_{Th} is threshold voltage, C is the capacitance, that is $1.05 \times 10^{-8} \text{ Fcm}^{-2}$ obtained from parallel plate model by considering $h\text{BN}$ and SiO_2 dielectrics. The maximum obtained n_G value is $4 \times 10^{12} \text{ cm}^{-2}$ at $-50 V_G$, that indicates a very small doping concentration induced by the gate dielectrics in BP channel. It is highly important to note here that, the Hall carrier density (n_H) at very high electrical field in Fig. 3(d) is several times higher than gate induced carriers in BP channel. It is, therefore, confirmed that the additional charge carriers inside the BP channel are due to the CM by impact ionization by hot carriers

By varying number of above parameters, we always observe super-linear I-Vs as shown in Fig. 1 and Supporting information S1 and S6~S8.

2) The thickness of the layers, both $h\text{BN}$ and BP should be given. These are important parameters to determine before one can understand and reproduce the electrostatics of the devices.

Reply:

We thank the reviewer for the comment. The detailed thickness information of our representative BP devices, whose data sets are shown in manuscript, is given in supporting information S4. Where we have compiled a table to enlist the thickness information of measured BP devices, as;

Device #	Geometry	Thickness (nm)			Remarks
		Bottom h BN	BP	Top h BN	
1	h BN/BP/ h BN	26	24	20	Two-probe device; Data set shown in Fig. 1 (b) and (c) for negative V_D breakdown
2	h BN/BP/ h BN	30	36	25	Two-probe device; Data set shown in Fig. 1 (d) and S5 for positive V_D breakdown.
3	h BN/BP/ h BN	15	28	9	Hall bar device; Data set shown in Fig. 3(c) and (d)
4	h BN/BP/ h BN	40	30	18	Hall bar device; Data set shown in Fig. 4 (a) and (b)
5	h BN/BP	25	6	-	Two terminal device; Data set shown in S7 for different dielectric comparison.
6	SiO ₂ /BP	285	33	-	Two terminal devices; Data set shown in S8 for anisotropic breakdown comparison.

As an example, the optical microscopic (OM) image and AFM images of two terminal *h*BN encapsulated BP device (device #1) is shown in following figures. The height profile of individual flake scanned by AFM is shown in given plots, wherein, *h*BN top, BP channel and *h*BN bottom are 20 nm, 24 nm and 26 nm thick, as indicated by black, red and blue colors respectively.

We have provided the thickness information of all the measured devices in the revised manuscript on page 6, line 7 from bottom, as;

“We repeated the experiments to the SiO₂ and/or hBN supported back gated BP devices of more than 30 different thicknesses (6 ~ 42 nm thick BP flakes)”

In captions of Fig. 3(b) of manuscript as;

“The thickness of hBN top, BP and hBN bottom is 9, 28, 15 nm respectively.”

Minor points:

1) On p. 5 is described the Raman spectra. The term “atomic oscillations” to describe BP vibrations is poorly chosen and probably wrong.

Reply:

We thank the reviewer for the comment, the above term is replaced with “atomic vibration” and that paragraph is moved to supporting information S3.

2) Some references are wrong or poorly selected (e.g. environmental sensitivity ascribed to Ref. 16 and 17), which makes me wonder if the authors have a good knowledge of the literature on both BP and FETs.

Reply:

We thank the reviewer for the comment. The above references were mistakenly placed there. We have replaced (ref. 16 and 17) with appropriate references (ref. 19 and 20) in revised manuscript.

The ref. 19 [Castellanos-Gomez, *et al.*, *2D Mater.* **1**, 25001 (2014)] explained the environmental degradation of BP flakes, while ref. 20 [Doganov *et al.*, *Nat. Commun.* **6**, 6647 (2014)] explained passivation of BP flakes by hBN.

Reviewer #3 (Remarks to the Author):

In this manuscript, the authors studied the impact ionization in black phosphorus. The data are coherent and support the conclusions well. For these reasons, I suggest a minor revision of the paper before it can be accepted by Nature Communications. Following are the questions that need to be addressed:

Reply:

We thank the reviewer for a careful reading of our manuscript and especially for his / her appreciation of the importance and novelty of our work in terms of fundamental physics and industry application potential. We truly appreciate several very important comments, we revised manuscript which fully addressed reviewer’s concerns.

1. On Page 6, the authors mention that “an alternative carrier transport mechanism is present and capable of masking current saturation in BP.” Can the authors explain what the carrier transport mechanism is?

Reply:

We thank the reviewer for the comment. The carrier transport mechanisms in above sentences points towards contact dominated transport, hopping transport and impact ionization. These three transport mechanisms were explained in subsequent paragraphs with details. Therefore, we revised the above sentence in manuscript as,

“an alternative carrier transport mechanism is present and capable of masking current saturation in BP, as explained subsequently.”

2. On Page 9, the authors mention “the effective mass of the holes is smaller than that of the electrons”. Can the authors indicate the effective mass of electrons and holes in BP?

Reply:

We thank the reviewer for the comments. The effective mass of holes and electrons of BP is given along both the in-plane directions on page 9, as;

“In the case of BP, the effective mass of the holes ($m_x = 0.076m_0$ and $m_y = 0.648m_0$) is smaller than that of the electrons ($m_x = 0.0826m_0$ and $m_y = 1.027m_0$) along both of the in-plane directions.” [Liu *et al.*, *Chem. Soc. Rev.* **44**, 2732–2743 (2015)]

3. What is the thickness of BP in the device used in the paper? More material characterization is needed.

Reply:

We thank the reviewer for the comment. The detailed thickness information of our representative BP devices, whose data sets are shown in manuscript, is given in supporting information S4. Where we have compiled a table to enlist the thickness information of measured BP devices, as;

Device #	Geometry	Thickness (nm)			Remarks
		Bottom hBN	BP	Top hBN	
1	hBN/BP/hBN	26	24	20	Two-probe device; Data set shown in Fig. 1 (b) and (c) for negative V_D breakdown
2	hBN/BP/hBN	30	36	25	Two-probe device; Data set shown in Fig. 1 (d) and S5 for positive V_D breakdown.
3	hBN/BP/hBN	15	28	9	Hall bar device; Data set shown in Fig. 3(c) and (d)
4	hBN/BP/hBN	40	30	18	Hall bar device; Data set shown in Fig. 4 (a) and (b)

5	h BN/BP	25	6	-	Two terminal device; Data set shown in S7 for different dielectric comparison.
6	SiO ₂ /BP	285	33	-	Two terminal devices; Data set shown in S8 for anisotropic breakdown comparison.

As an example, the optical microscopic (OM) image and AFM images of two terminal *h*BN encapsulated BP device (device #1) is shown in following figures. The height profile of individual flake scanned by AFM is shown in given plots, wherein, *h*BN top, BP channel and *h*BN bottom are 20 nm, 24 nm and 26 nm thick, as indicated by black, red and blue colors respectively.

We have provided the thickness information of all the measured devices in the revised manuscript on page 6, line 7 from bottom, as;

“We repeated the experiments to the SiO₂ and/or *h*BN supported back gated BP devices of more than 30 different thicknesses (6 ~ 42 nm thick BP flakes)”

In captions of Fig. 3(b) of manuscript as;

“The thickness of *h*BN top, BP and *h*BN bottom is 9, 28, 15 nm respectively.”

Reviewers' comments:

Reviewer #1 (Remarks to the Author):

The issues that I brought up were thoughtfully addressed with additional experiments, new shreds of evidence, and clarification of device details. In my opinion, the current revised version is OK to be published in Nature Communication.

Reviewer #2 (Remarks to the Author):

I am satisfied with most of the answers provided by the authors to my concerns and questions. They have corrected the paper and included new elements such as screening conditions and device parameters. However, I feel that the discussion on the conditions for CM is a bit shallow. The authors have explored a set of screening conditions, but the variations in thicknesses are not quite sufficient, in my view, to support a definitive conclusion on CM. I understand that it would be difficult to experimentally push the gate oxide and channel thickness down further with these BP devices towards the expected saturation regime, but in the absence of a systematic exploration of the device parameters, one can still think that short channel effects rather than CM are behind the observed super-linear IVs. Nevertheless, I feel that the study is interesting and I would support publication provided that the authors address the remaining points below:

- 1) The authors should weaken their claims about CM or add a discussion ruling out possible artefacts such as behaviours due to short channel FETs (due to a poor scaling of the device).
- 2) Raman is performed on the layers, yet there is no analysis from those results about the quality of the layer (e.g. estimate of A_{1g}/A_{2g} ratio compared to that expected for pure layers, as explained in Nat. Mat. 14, 826 (2015)). It is important to know that the layers measured are not extensively damaged and the presence of a protecting layer is no proof of having no degradation.

Reviewers' comments:

Reviewer #2 (Remarks to the Author):

I am satisfied with most of the answers provided by the authors to my concerns and questions. They have corrected the paper and included new elements such as screening conditions and device parameters. However, I feel that the discussion on the conditions for CM is a bit shallow. The authors have explored a set of screening conditions, but the variations in thicknesses are not quite sufficient, in my view, to support a definitive conclusion on CM. I understand that it would be difficult to experimentally push the gate oxide and channel thickness down further with these BP devices towards the expected saturation regime, but in the absence of a systematic exploration of the device parameters, one can still think that short channel effects rather than CM are behind the observed super-linear IVs. Nevertheless, I feel that the study is interesting, and I would support publication provided that the authors address the remaining points below:

1) The authors should weaken their claims about CM or add a discussion ruling out possible artefacts such as behaviors due to short channel FETs (due to a poor scaling of the device).

Reply:

We thank the reviewer for the comment. We agree with the reviewer's comment that the short channel effect should be ruled out before claiming the CM as a responsible mechanism. In the original manuscript, we have discussed several other possible mechanisms such as large contact resistance and hopping transport that are duly ruled out based on experimental results and technical discussion. Furthermore, in the revised manuscript, we have added additional discussion about short channel effect.

It has been reported that the semiconducting 2D materials exhibit immunity to the short channel effect thanks to their low dielectric constant and ultra-thin body (Chhowalla *et. al.*, Nat. Rev. Mater., 1, 16052, 2016 and Liu *et. al.*, ACS Nano, 6, 8563-8569, 2012). Furthermore, layered semiconducting BP exhibits a little sign of short channel effect only if its channel length is scaled to 20 nm (Miao *et. al.*, ACS Nano, 9, 9236-9243, 2015). Here, in our study, the smallest measured device has the channel length of 2 μm , that is two orders longer than that of used in the aforementioned study. Based on this, we think it is quite unlikely to observe short channel effect in our measured BP devices.

The revision is carried out at page 8, line 9 – 613, as;

“Besides these, there might be possibility of other artifacts such as short channel effect in our BP devices. Miao *et al.* observed a weak short channel effect in the ultra-narrow channel (20 nm) BP devices²⁶. Ours measured shortest channel BP device is over two

orders longer ($> 2 \mu\text{m}$) than observed short channel BP devices, and thereby, we rule out the possibility of short channel effect.”

2) Raman is performed on the layers, yet there is no analysis from those results about the quality of the layer (e.g. estimate of A_{1g}/A_{2g} ratio compared to that expected for pure layers, as explained in *Nat. Mat.* 14, 826 (2015)). It is important to know that the layers measured are not extensively damaged and the presence of a protecting layer is no proof of having no degradation.

Reply:

We thank the reviewer for the comment. The environmental degradation is one of the most important issue for black phosphorus. To protect BP from external perturbation, we encapsulated it with *h*BN, that has already been reported as an effective passivation of BP (Dogonov et. al., *Nat. Commun.* **6**, 6647 (2015)). For further proof, we carried out aging test of BP flake based on Raman spectroscopy. The obtained results are added in Supporting Information S11, as;

“2.11 Aging test of BP flakes

BP is notorious for its environmental sensitivity, and therefore, we performed aging test to monitor the degradation of BP and the effectiveness of *h*BN capping. We exfoliated few layer BP flake ($7 \sim 8 \text{ nm}$) on SiO_2 , and partially covered it with 25 nm thick *h*BN flake inside the glove box, as shown in Fig. S11(a). Surprisingly, the naked BP flake degraded so fast that it completely disappeared just after twelve hours of high humid (75%) ambient exposure, as shown in Fig. S11(b). This is further confirmed by spatial Raman intensity mapping (A_{2g} mode of BP) that witnesses the vanishing of naked BP flake, while BP flake still exists underneath *h*BN, as shown in Fig. S11(c). For more details, see temporal sequence of optical microscope images from (i - vii) in Fig. S11(d).

FIG.S11 Optical microscope (OM) images of bare and *h*BN passivated BP flakes with time. (a) OM image of targeted flakes soon after exfoliation and transfer in glove box where red shaded boundary region denotes few layers BP flake partially covered with *h*BN, and blue shaded region is *h*BN. Scale bar is 5 μm . (b) OM image of same flakes after 720 minutes exposure to ambient conditions. The exposed BP flake disappeared in that span of time. (c) Spatial Raman intensity map (A^2_g mode of BP) of red rectangle-indicated region shown in (b), confirming existence of BP under *h*BN, here scale bar is 3 μm . (d) False color OM images taken after given time span from i - vii.

Raman spectroscopy is a non-invasive and highly effective technique to characterize the material quality¹². Therefore, we acquired time dependent Raman characteristics of exposed and *h*BN passivated BP flake to monitor its degradation in the ambient conditions. From our obtained results, we plotted the ratio of deconvoluted A^1_g and A^2_g modes of BP against time, see Fig. S12. Interestingly, a decreasing trend is

observed for the bare BP flake, that is the signature of oxidations. However, the passivated BP flake shows stubborn spectra, thanks to effective passivation of BP by *h*BN.

FIG.S12 **Time dependent Raman spectra of BP.** Raman intensity peak ratio (A_g^1/A_g^2) as a function of time collected from bare and *h*BN passivated BP flake.

REVIEWERS' COMMENTS:

Reviewer #2 (Remarks to the Author):

I have no more concern about the paper and the manuscript can now be published as is.